# Position: Agents Should Invoke External Tools ONLY When Epistemically Necessary

**Hongru Wang** [1 2]  **Cheng Qian** [3]  **Manling Li** [4]  **Jiahao Qiu** [5]  **Boyang Xue** [2]
**Mengdi Wang** [5]  **Heng Ji** [3]  **Amos Storkey** [1]  **Kam-Fai Wong** [2]

🌐 **Welcome to Theory of Agent!**     📖 **Notion Blog**

## Abstract

As large language models evolve into tool-augmented agents, a central question remains unresolved: when is external tool use actually justified? Existing agent frameworks typically treat tools as ordinary actions and optimize for task success or reward, offering little principled distinction between epistemically necessary interaction and unnecessary delegation. This position paper argues that *agents should invoke external tools only when epistemically necessary*. Here, epistemic necessity means that a task cannot be completed reliably via the agent's internal reasoning over its current context, without any external interaction. We introduce the ***Theory of Agent (ToA)***, a unified framework that reconceives agents as *tool-use decision makers* under epistemic uncertainty to decide whether remaining uncertainty should be resolved internally or delegated externally. From this perspective, common agent failure modes (e.g., overthinking and overacting) arise from miscalibrated decisions under uncertainty rather than deficiencies in reasoning or tool execution alone. We further discuss implications for training, evaluation, and agent design, highlighting that unnecessary delegation not only causes inefficiency but can impede the development of internal reasoning capability. Our position provides a normative and fundamental criterion for tool use that complements existing decision-theoretic models and is essential for building agents that are not only correct, but increasingly intelligent.

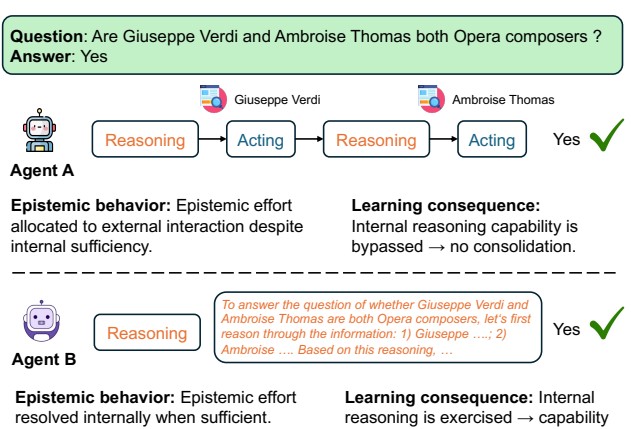

*Figure 1.* Tool-use decisions shape the trajectory of agent intelligence. Two agents may achieve comparable task success through different allocations of epistemic effort. An over-delegating agent frequently invokes external tools even when internal reasoning suffices, resulting in stagnant internal capability despite correctness. In contrast, an epistemically calibrated agent invokes external tools only when necessary, allowing internal reasoning capability to expand over time as experience accumulates. This figure illustrates our central position: external tools should be invoked only when epistemically necessary, since unnecessary delegation reshapes not just efficiency, but the trajectory of agent intelligence itself. The example is drawn from Wang et al. (2025a).

## 1. Introduction

Large Language Models (LLMs) have rapidly evolved beyond text generation into autonomous agents capable of independently planning and executing complex tasks with minimal human oversight (Kolt, 2025). These emerging capabilities have enabled a broad range of real-world applications, including travel planning (Xie et al., 2024a), human-computer interaction (Xie et al., 2024b; Wang et al., 2024b; Qin et al., 2025), and scientific research (Nguyen et al., 2024; Edwards et al., 2024). As a result, a central question in agent design has emerged: *when should an agent rely on its internal reasoning (e.g, continue existing reasoning processing over the context), and when should it interact with the external world to gain new information?*

This paper takes a clear position: **an agent should invoke external tools only when epistemically necessary**. That is,

---

[1]University of Edinburgh [2]The Chinese University of Hong Kong [3]University of Illinois Urbana-Champaign [4]Northwestern University [5]Princeton University. Correspondence to: Hongru Wang <hongru.carrywang@gmail.com>, Cheng Qian <chengq9@illinois.edu>.

*Proceedings of the 43rd International Conference on Machine Learning*, Seoul, South Korea. PMLR 306, 2026. Copyright 2026 by the author(s).

external interaction is justified *if and only if* the remaining uncertainty required to complete a task cannot be resolved through internal reasoning alone. Despite the prevalence of tool-augmented agents, existing paradigms lack a principled criterion for this decision. Agents are often trained to maximize task success, minimize cost, or follow predefined workflows. These approaches permit agents to overuse tools, steps, or reasoning tokens as long as answers are correct, treating tool use as a free shortcut rather than an epistemic commitment. As a consequence, agents exhibit common failure modes—such as overthinking (Cuadron et al., 2025; Huang et al., 2026) and overacting (Qian et al., 2025; Wang et al., 2025a)—that are poorly explained as isolated reasoning or execution errors. We argue that these behaviors are best understood as failures of sequential decision making under the unified epistemic uncertainty. At each step of execution, an agent must decide how to allocate epistemic effort: whether to resolve uncertainty internally through reasoning, or to delegate it externally through interaction. This decision cannot eliminate epistemic difficulty; it can only reallocate where that difficulty is resolved. Crucially, unnecessary delegation does more than introduce inefficiency—it suppresses the development of internal reasoning capability, shaping the long-term intelligence of the agent, as shown in Figure 1.

To formalize this view, we systematically introduce the ***Theory of Agent (ToA)***, a unified framework that reconceives agents as *tool-use decision makers under epistemic uncertainty*, built on three contributions: (1) a unified agent framework treating reasoning and acting as co-equal knowledge-acquisition tools; (2) a formal knowledge boundary that separates the internal task set (tasks an agent can solve through internal reasoning alone given current context) from the world task set (tasks that require external interaction), together with population-relative extensions capturing what is solvable across a family of agents. Since this boundary is latent, agents must approximate it through belief-based solvability estimates and policy-dependent thresholds; and (3) several normative propositions that convert the framework into practical guidance, centered on the notion of ***epistemic effort***: an invariant task requirement that no policy can eliminate, only redistribute between internal reasoning and external interaction. Within this view, alignment is not defined by correctness alone but by effort-consistent decision making: allocating epistemic effort in a manner consistent with the agent's internal capabilities and its long-term development. This perspective yields concrete implications for evaluation, revealing why correctness achieved through excessive tool use is still misaligned, and for learning, explaining why systems that over-delegate stagnate in internal reasoning capability.

The remainder of the paper develops this position in a stepwise manner. Section 2 reframes reasoning and acting as alternative means of knowledge acquisition, providing a unified foundation for analyzing tool-use decisions. Section 3 introduces the core theoretical objects of ToA, including internal and world task sets, knowledge boundaries, and epistemic effort-and shows how common agent failures arise from misaligned effort allocation rather than insufficient capability. Section 4 then examines how these principles shape learning and training dynamics, explaining why unnecessary delegation can stall internal reasoning development even when task performance remains high. Section 5 contrasts this epistemic view with prevailing agent paradigms that optimize for correctness, reward, or workflow execution. We conclude by arguing that tool-use decisions are not merely an efficiency concern, but a defining factor in the long-term trajectory of agent intelligence.

## 2. Foundations

### 2.1. Reasoning and Acting as Alternative Means of Knowledge Acquisition

Intelligent agent behavior is commonly described in terms of two capabilities: reasoning and acting (Yao et al., 2023; Wang et al., 2024c). Reasoning allows an agent to infer, plan, reflect, and manipulate information already available to it, while acting allows the agent to intervene in or query the external environment to obtain new information or cause state changes. In most agent frameworks, these capabilities are treated as qualitatively different stages (e.g., reasoning first, action later), or as loosely coupled components in a pipeline. We argue that this separation obscures a more fundamental question faced by autonomous agents: *how should an agent decide whether remaining uncertainty should be resolved internally or through external interaction?* To make this decision explicit, we adopt a unifying abstraction in which both reasoning and acting are treated as forms of tool use for knowledge acquisition.

> **Position: Reasoning and Acting as Knowledge-Acquisition Tools**
>
> Reasoning and acting are treated as alternative tools for reducing epistemic uncertainty: reasoning entails an internal cognitive tool that operate exclusively over information already available within the agent (e.g., parameters and context), while acting entails an external physical tools acquire new information or induce state changes through interaction with the environment.

This unification is not a claim that reasoning and acting are identical, nor a restatement of standard agent pipelines. Rather, it reframes both as *decision alternatives* for resolving uncertainty. The distinction between them lies in the provenance of information they access: internal tools reorganize or recombine existing information, whereas external tools introduce new information or effects that were not

previously available to the agent.

Under this view, *internal cognitive tools* refer to internal cognitive mechanisms that support systematic or investigative thinking to solve problems (Jonassen, 1992; Kommers et al., 1992), while *external physical tools* refer to modules or interfaces outside the model that are invoked through specific triggers, such as rules, actions, or special tokens, whose outputs are then incorporated into the model's context to inform subsequent reasoning (Hao et al., 2023; Lu et al., 2025). As a result, internal reasoning can fail when required information lies outside this scope, leading to fail trajectories. Conversely, external interaction is not intrinsically superior or more reliable. When agents invoke external tools despite sufficient internal information, they introduce unnecessary interaction, latency, and dependency on the environment. More importantly, such delegation bypasses opportunities for internal knowledge consolidation and reasoning development [1].

Treating reasoning and acting as alternative, co-available tools clarifies that neither should be privileged a priori. They are *co-equal* in the precise sense that each is justified only insofar as it can reduce remaining epistemic uncertainty. This perspective goes beyond the status quo by making tool-use decisions explicit objects of analysis. Rather than optimizing reasoning quality or action efficiency in isolation, it enables principled evaluation of when interaction is epistemically necessary and when it constitutes unnecessary delegation, setting the stage for the normative position developed in Section 3.

## 2.2. Tool-Integrated Agents

We follow the general POMDP formulation of agents (Åström, 1965; Christianos et al., 2023; Narayanan et al., 2024), in which an agent is a policy $\pi$ mapping histories to actions, and overlay an epistemic annotation of the action space that makes tool-use decisions explicit. Formally, let an agent with model $m$ interact with the environment $\mathcal{W}$ over discrete steps. At step $t$m the agent observes the history:

$$\tau_t \triangleq (q, a_1, o_1, \ldots, a_{t-1}, o_{t-1}),$$

where $q$ specifies the task, and selects an action or tool $a_t \sim \pi(\cdot \mid \tau_t)$ from an action set $\mathcal{T}$, receiving observation $o_t$ in return. All of this is standard POMDP machinery. On top of this formalism, we partition $\mathcal{T}$ by the information provenance into of each action:

$$\mathcal{T} = \mathcal{T}_{\text{int}} \cup \mathcal{T}_{\text{ext}},$$

where $\mathcal{T}_{\text{int}}$ denotes *internal cognitive tools* (e.g., reasoning, reflection), and $\mathcal{T}_{\text{ext}}$ denotes *external physical tools* (e.g.,

---

[1]We provide more concrete examples for internal cognitive tool and external physical tool in Appendix A.

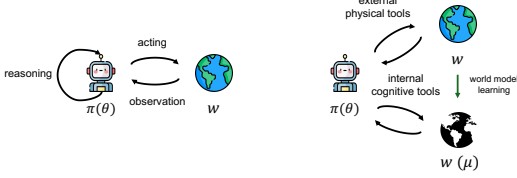

(a) Previous: ReAct-based Agent    (b) Ours: Tool-integrated Agent

*Figure 2.* An unified agent framework that integrates both internal cognitive tools and external physical tools with agent itself as a world model.

search, APIs, UI actions). These two classes differ in a structurally important way:

- ***Internal tools query the internal world model.*** Their "observations" are outputs generated by the agent's own model, $o_t = f_m(\tau_t, a_t)$, which extend the agent's context $\tau_t$ without changing the environment state $s_t$.

- ***External tools query the external physical world.*** Their observations are drawn from the external environments, reflect the true environment state $s_t$ and may change it.

Equivalently, the transition can be written as:

$$(s_{t+1}, o_t) = \begin{cases} (s_t, \ f_m(\tau_t, a_t)) & a_t \in \mathcal{T}_{\text{int}}, \\ (s_{t+1} \sim P, \ o_t \sim O(\cdot \mid s_t, a_t)) & a_t \in \mathcal{T}_{\text{ext}}. \end{cases}$$
$$(1)$$

This unified framework offers several key advantages: (1) It is compatible with existing theoretical formulation by encoding this distinction as an annotation on the action space, rather than as a modification of the POMDP itself, preserves the standard formalism while making tool-use decisions a first-class object of analysis; (2) It generalizes prior practical approaches such as ReAct (Yao et al., 2023), which can be viewed as special cases where internal tool steps (e.g., reasoning) are treated as monolithic thought units $r_i$, leveraging the model's pre-trained cognitive abilities without requiring explicit tool separation; (3) It aligns with findings from large reasoning models (LRMs), which show that outcome-based reinforcement learning (RL) can effectively train agents to discover and utilize internal cognitive tools (DeepSeek-AI, 2025). The same principle applies to external physical tools, as shown in recent studies on tool-augmented agents (Jin et al., 2025). Thus, the framework provides a coherent foundation for agentic learning across both domains. (4) Most importantly, this perspective shed lights to a potential unified path combining agent learning and world modeling (Richens et al., 2025; Li et al., 2026), as shown in Fig 2.

# 3. Position: Agent Should Invoke External Tools ONLY When Epistemically Necessary

Autonomous agents based on large language models repeatedly face a fundamental decision during task execution: *Should I continue reasoning internally, or should I interact with the external world to obtain more information?* This decision arises at every step of agent execution and underlies common failure modes such as overthinking and overacting. We argue that these behaviors are best understood as failures in sequential decision-making under epistemic uncertainty, rather than as isolated deficiencies of reasoning or tool execution. In this section, we formalize agent behavior from this perspective and articulate a small set of propositions that together characterize effective tool-use decisions.

## 3.1. Task Sets and Knowledge Boundary

We now define the core ontological object in our framework.

**Definition 3.1** (Internal and World Task Sets). Let $\mathcal{Q}$ denote the space of solvable tasks defined by the environment. For a given fixed agent [2] or model $m$ and environment $\mathcal{W}$, define:

- The *internal task set* $\mathcal{Q}^{\mathrm{int}}(m, \mathcal{W})$ as the set of tasks that the agent can complete reliably using internal reasoning alone, without invoking external tools.

- The *world task set* $\mathcal{Q}^{\mathrm{world}}(\mathcal{W})$ as the set of all tasks that are in principle solvable in the environment given access to appropriate external interaction. By construction, $\mathcal{Q}^{\mathrm{int}}(m, \mathcal{W}) \subseteq \mathcal{Q}^{\mathrm{world}}(\mathcal{W})$

**Definition 3.2** (Knowledge Boundary). The *knowledge boundary* of an agent model $m$ is defined by the separation between its internal task set $\mathcal{Q}^{\mathrm{int}}(m, \mathcal{W})$ and the world task set $\mathcal{Q}^{\mathrm{world}}(\mathcal{W})$. Tasks in $\mathcal{Q}^{\mathrm{ext}}(m, \mathcal{W}) = \mathcal{Q}^{\mathrm{world}}(\mathcal{W}) \setminus \mathcal{Q}^{\mathrm{int}}(m, \mathcal{W})$ require new external interaction to be completed reliably by the agent. Please note this does not mean the agent can not use existing external tool for internal tasks.

We further discuss how the knowledge boundary evolves over time — and the distinct regimes that arise in static versus dynamic worlds — in Appendix B.4.

**Observation 1: Model-Specific Knowledge Boundaries.** For a given environment $\mathcal{W}$, different agent models may exhibit different internal task sets $\mathcal{Q}^{\mathrm{int}}(m, \mathcal{W})$ due to differences in training data, architecture, memory, and available tools. In sufficiently simple environments, these sets may coincide across models. However, in realistic or challenging task regimes, the location of the knowledge boundary is model-dependent, implying that no single tool-use policy can be uniformly appropriate across all agents. We provide a illustration in Fig 3.

---

[2]A fixed agent means the memory and tools are also fixed.

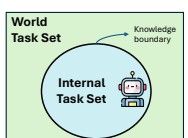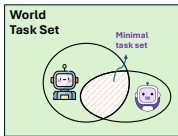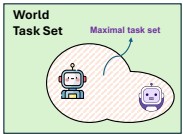

*Figure 3.* The high-level illustration of internal, world and population-relative task sets.

**Definition 3.3** (Population-relative Task Sets). Let $\mathcal{M}_{\mathrm{agents}}$ denote a fixed population of agent models. We define minimal and maximal task set as:

$$\mathcal{Q}_{\min} \triangleq \bigcap_{m \in \mathcal{M}_{\mathrm{agents}}} \mathcal{Q}^{\mathrm{int}}(m), \tag{2}$$

$$\mathcal{Q}_{\max} \triangleq \bigcup_{m \in \mathcal{M}_{\mathrm{agents}}} \mathcal{Q}^{\mathrm{int}}(m). \tag{3}$$

$\mathcal{Q}_{\min}$ captures competence shared by all agents in the population, while $\mathcal{Q}_{\max}$ represents the envelope of internal competence achievable within the population [3].

**Proposition 3.4** (Population-Relative Tool Necessity). *For a fixed agent population $\mathcal{M}_{\mathrm{agents}}$, any task*

$$q \in \mathcal{Q}^{\mathrm{world}} \setminus \mathcal{Q}_{\max}$$

*cannot be solved internally by any agent in the population. Consequently, successful completion of such tasks requires external interaction for all agents in $\mathcal{M}_{\mathrm{agents}}$.*

## 3.2. Tool Use as Belief-Based Classification

In practice, $\mathcal{Q}^{\mathrm{int}}(m)$ is latent and cannot be observed directly. Therefore, it is important make tool-use decisions under epistemic uncertainty, based on the agent's internal estimates of whether a task lies within its internal task set (e.g., self-awareness). As a result, tool-use behavior reflects a stochastic and context-dependent classification of tasks relative to the knowledge boundary, rather than a fixed decision boundary.

**Definition 3.5** (Context-Conditioned Internal Solvability (Operational Estimate)). For a task $q$, agent model $m$, environment $\mathcal{W}$, and interaction context $\tau_t$, define

$$p_t^{\mathrm{int}}(q, m; \mathcal{W}) \triangleq \Pr(S = 1 \mid q, m, \tau_t, \mathcal{W}, \pi \in \Pi_{\mathrm{int}}) \tag{4}$$

where $\Pi_{\mathrm{int}}$ denotes policies that do not invoke external tools and $S = 1$ indicates successful task completion.

It is a belief-like operational surrogate used by the agent to estimate whether the current task lies within its internal task set given the available context. In practice, there are several ways to approximate such value such as self-consistency

---

[3]Throughout this subsection, we assume a fixed environment $\mathcal{W}$ and omit it from notation when unambiguous.

rollout (Wang et al., 2023b), draft-reasoning confidence (Xu et al., 2024; Wang et al., 2025b), hidden-state probes (Bu et al., 2026) and so on.

**Definition 3.6** (Operational Threshold)**.** Let $\alpha \in (0,1)$ denote a policy-dependent confidence threshold. An agent treats a task $q$ as internally solvable at step $t$ if $p_t^{\text{int}}(q, m; \mathcal{W}) \geq \alpha$.

**Observation 2: Belief-Dependent Classification.** Different agents, or the same agent at different stages of interaction, may assign different values of $p_t^{\text{int}}(q, m; \mathcal{W})$ to the same task. Moreover, even given identical estimates, agents may exhibit different tool-use behavior due to different policy thresholds $\alpha$, reflecting differences in risk tolerance, efficiency preferences, or deployment constraints.

**From Belief to Action.** Given an operational estimate $p_t^{\text{int}}(q, m; \mathcal{W})$ and a policy-dependent threshold $\alpha$, a natural class of tool-use policies can be expressed as belief-conditioned decision rules. For example, an agent may choose to rely on internal reasoning when $p_t^{\text{int}}(q, m; \mathcal{W}) \geq \alpha$, and invoke external tools otherwise. Crucially, this rule does not define the agent's true knowledge boundary, but only specifies how epistemic beliefs are translated into actions under uncertainty. Different agents may adopt different thresholds or more complex mappings, leading to diverse tool-use behaviors even when underlying task sets coincide.

Although internal solvability is latent, it is not static. As agents reason or interact, the available context typically expands, incorporating intermediate results, partial conclusions, or newly acquired information. For tasks where such context is relevant, this expansion can strictly increase the agent's internal solvability estimate, effectively shifting the knowledge boundary during execution. We formalize this monotonicity property as follows:

**Proposition 3.7** (Context Expansion Tends to Increase Internal Solvability)**.** *Let $\tau_t \subseteq \tau_{t'}$ for $t' > t$ denote that the available context is expanded. For tasks $q$ for which the additional context is relevant and non-degrading,*

$$p_t^{\text{int}}(q, m; \mathcal{W}) \leq p_{t'}^{\text{int}}(q, m; \mathcal{W}), \quad \text{and hence} \quad (5)$$
$$\mathcal{Q}_t^{\text{int}}(m; \mathcal{W}) \subseteq \mathcal{Q}_{t'}^{\text{int}}(m; \mathcal{W}). \quad (6)$$

This proposition formalizes the intuition that, as interaction proceeds, agents may accumulate relevant intermediate results, clarifications, or partial progress in $\tau_t$ that enables more tasks to be solved internally.

### 3.3. Effort Allocation under Epistemic Uncertainty

While task sets and knowledge boundaries define what an agent *can* solve internally, they do not determine *how* an agent should allocate internal reasoning and external interac-

tion during execution. To capture this, we introduce the notion of *epistemic effort*, which captures the task-dependent informational burden that must be resolved for successful completion. Importantly, epistemic effort is not a measure of cost or efficiency, but a structural requirement imposed by the task relative to the agent's capabilities.

An agent may satisfy this requirement through a combination of internal reasoning and external interaction. We express this decomposition as

$$E(q, m) = E_{\text{int}}(q, m) + E_{\text{ext}}(q, m), \quad (7)$$

where $E_{\text{int}}(q, m)$ denotes effort satisfied through internal cognitive tools and $E_{\text{ext}}(q, m)$ denotes effort satisfied through external physical tools.

**Proposition 3.8** (Epistemic Effort as an Unavoidable Requirement)**.** *Fix an agent $m$, environment $\mathcal{W}$, and task $q \in \mathcal{Q}^{\text{world}}(\mathcal{W})$. Let $\Pi_{\text{succ}}(q, m, \mathcal{W})$ denote the set of policies that successfully complete $q$. Define the task's minimal required epistemic effort as the infimum:*

$$E^{\star}(q, m) \triangleq \inf_{\pi \in \Pi_{\text{succ}}(q, m, \mathcal{W})} \left( E_{\text{int}}^{\pi}(q, m) + E_{\text{ext}}^{\pi}(q, m) \right). \quad (8)$$

*Then for any successful policy $\pi \in \Pi_{\text{succ}}(q, m, \mathcal{W})$,*

$$E_{\text{int}}^{\pi}(q, m) + E_{\text{ext}}^{\pi}(q, m) \geq E^{\star}(q, m). \quad (9)$$

*By definition, no successful policy can eliminate this requirement; it can only reallocate effort between internal reasoning and external interaction.*

**Interpretation.** This proposition formalizes a fundamental constraint on agent behavior. External tools do not remove epistemic difficulty; they shift where the difficulty is resolved. Stronger agents—those with larger internal task sets—can satisfy a greater fraction of $E^{\star}(q, m)$ internally, while weaker agents rely more heavily on external interaction. However, the total epistemic effort required by the task is invariant to the agent's strategy. This invariance explains why agents with different internal capabilities may achieve similar task performance through different effort allocations, and why excessive internal reasoning or excessive tool use both represent inefficient responses to epistemic uncertainty. We provide a high-level illustration in Fig 4.

We emphasize that there are several additional important features shown in the figure: 1) By definition, all points at the segment $AC$ and $AB$ are ideal allocation, but the optimal objective depends on different preferences in terms of safety, speed, time and other issues. Our normative concern is unintentional drift toward $A$ (or beyond) from correctness-only training, suppressing internal capability development; 2) The optimal effort decomposition extends to graded costs:

$$E_{total}^{*} = c_{int} \cdot E_{int} + c_{ext} \cdot E_{ext} \quad (10)$$

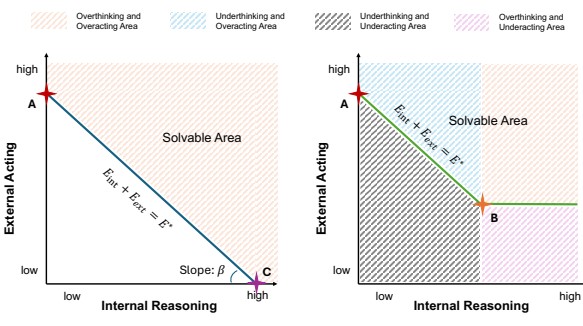

**(a) Epistemic Effort for Internal Task Set**    **(b) Epistemic Effort for External Task Set**

*Figure 4.* A high-level illustration of epistemic effort allocation and failure modes in tool-augmented agents for: (a) task in $\mathcal{Q}^{\text{int}}(m, \mathcal{W})$; and (b) task in $\mathcal{Q}^{\text{ext}}(m, \mathcal{W})$. The difference is the internal and external efforts can transfer freely for internal task (e.g., the segment $AC$), but external efforts are necessary for external task. **Point A** represents pure delegation in which the agent immediately calls an external tool (e.g., a stronger agent) with negligible internal reasoning. **Point B** denotes the ideal frontier, corresponding to the minimal external effort required to solve the task when internal reasoning is fully exploited. **Point C** represents pure internal reasoning without any external interaction. The shaded regions illustrate characteristic misallocations of effort, such as underthinking, overacting, and their combinations, arising from decision-boundary misalignment rather than insufficient capability. The solvable area represents the set of feasible internal–external effort allocations under which the agent can successfully solve a problem.

where $c_{int}$ captures per-token cost (quadratic attention, KV cache growth) and $c_{ext}$ captures per-call cost (API latency, monetary fees). When CoT cost is high relative to external call cost, rational delegation shifts the ideal operating point toward $A$ in Fig 4 — justified by resource rationality even without strict epistemic necessity. The $\alpha$ implicitly encodes $c_{int}/c_{ext}$: high internal cost $\to$ lower $\alpha$ is rational. The slope $\beta$ in Fig 4 exactly stands for this ratio and this connects to resource-bounded rationality. While these features are crucial for aligning future agents with different users and environments, we advocate that the ultimate objective, viewed purely from the standpoint of intelligence itself, is to converge to point $C$ for internally solvable tasks and point $B$ for externally required tasks; 3) Notably, point $A$ appears in both panels reveals why correctness-only training drifts toward delegation: $A$ remains a solvable strategy regardless of whether the task lie in $\mathcal{Q}^{int}$ or $\mathcal{Q}^{ext}$, whenever a sufficiently capable external agent is available. This task-agnostic feasibility makes $A$ an attractor under outcome-only objectives, but it also erases the very distinction between these tasks, which is why our normative ideal lies instead at $C$ and $B$: only these points preserve the signal that distinguished the two type of tasks and sustains internal capability development.

**Proposition 3.9** (Delegation-Induced Capability Stagnation). *If an agent systematically allocates epistemic effort to external interaction for tasks that lie within its internal*

*task set, then its internal reasoning capability for those tasks will not improve through experience, even when such improvement is possible in principle* [4].

**Relation to Autonomous Intelligence.** Our position aligns closely with LeCun's vision of autonomous machine intelligence (LeCun et al., 2022), which emphasizes minimizing real-world actions by internalizing knowledge about the world. Within our framework, this principle arises naturally: external actions are minimized when internal reasoning is sufficient to satisfy the remaining epistemic effort. Crucially, ToA makes explicit the decision mechanism underlying this behavior—external interaction is justified only when epistemically necessary—and shows that unnecessary delegation not only reduces efficiency, but also impedes the development of internal intelligence.

**Implications for Evaluation.** Although the internal task set $\mathcal{Q}^{\text{int}}(m, \mathcal{W})$ and the minimal epistemic effort $E^\star(q, m)$ are latent and cannot be directly observed, Proposition 3.8 provides a principled basis for evaluating agent behavior. Since epistemic effort cannot be eliminated, agents can only differ in how they allocate effort between internal reasoning and external interaction. As a result, evaluation should focus not solely on task success, but on patterns of effort reallocation: whether agents increasingly satisfy epistemic requirements internally as their competence grows, whether they invoke external tools only when internal reasoning is insufficient, and whether additional effort contributes meaningfully to task progress. Agents that achieve correctness while systematically over-allocating either internal or external effort can therefore be identified as miscalibrated, even when their final answers are correct.

### 3.4. Alignment as Effort-Consistent Decision Making

The preceding analysis motivates a central behavioral requirement for agents: tool-use decisions should allocate epistemic effort in a manner consistent with both immediate task requirements and long-term capability development. At each step of execution, an agent implicitly chooses how to allocate epistemic effort. Invoking internal reasoning allocates effort to $E_{\text{int}}$, while invoking external tools allocates effort to $E_{\text{ext}}$. Because the total required effort $E^\star(q, m)$ is fixed, misallocation does not reduce difficulty—it merely shifts inefficiency and alters learning dynamics.

**Effort-Consistent Alignment.** An agent is aligned if its tool-use decisions allocate epistemic effort in a manner consistent with its internal task set. Specifically, internal reasoning should be applied when it meaningfully reduces

---

[4]External tools can compensate for missing information, but they cannot replace the experience required to develop it. We provide detailed explanations in Appendix B.1.

epistemic uncertainty, and external interaction should be invoked only when internal reasoning is insufficient to do so. To make this concrete, consider an agent's internal solvability estimate $p_t^{\text{int}}(q)$ at step $t$. A simple decision rule illustrates the principle:

$$\pi(a_t \mid \tau_t) = \begin{cases} a_t \in \mathcal{T}_{\text{int}}, & p_t^{\text{int}}(q, m, \mathcal{W}) \geq \alpha, \\ a_t \in \mathcal{T}_{\text{ext}}, & p_t^{\text{int}}(q, m, \mathcal{W}) < \alpha, \end{cases} \quad (11)$$

where $\alpha$ is a policy-dependent operating threshold. Crucially, this rule is not optimal because it minimizes cost or latency, but because it respects the epistemic effort invariant: external tools are invoked only when internal effort cannot satisfy the remaining epistemic requirement.

**Proposition 3.10** (Effort-Consistent Decision Alignment). *Effective tool-use policies allocate epistemic effort such that internal reasoning is used for tasks within the agent's internal task set, while external interaction is used primarily for tasks outside it. Systematic deviation from this allocation leads to inefficiency without reducing the total epistemic effort required for task completion.*

**Corollary 3.11** (Failure Modes as Effort Misallocation). *Systematic overuse of internal reasoning allocates effort where it cannot reduce uncertainty, leading to overthinking and hallucination. Systematic overuse of external tools reallocates effort unnecessarily, leading to overacting and inefficiency. Both failure modes violate effort-consistent alignment, even when final task success is achieved.*

Together, these results show that alignment is not merely about producing correct outputs, but about allocating epistemic effort in a way that supports both reliable behavior and the continued development of internal intelligence.

# 4. Implications for Learning and Training

Until now, we establish that tool-use behavior is governed by epistemic effort allocation under uncertainty: agents cannot eliminate epistemic effort, only decide whether it is resolved internally or delegated externally. This section examines the consequences of this view for learning and training. We argue that effective agents require calibrated meta-cognition to allocate effort consistently with their internal capabilities (Section 4.1), and that misalignment not only causes inefficiency but also shapes long-term intelligence development. We analyze how this principle manifests during training and inference, characterize observable behavioral regimes (Section 4.2), and discuss practical pathways toward effort-consistent agent behavior (Section 4.3).

## 4.1. Meta-Cognition as Solvability Assessment

Given that epistemic effort must be allocated rather than eliminated, agents require a mechanism for deciding *where*

remaining uncertainty should be resolved. We argue that this mechanism is meta-cognition (Dunlosky & Metcalfe, 2008; Ackerman & Thompson, 2017): the agent's ability to assess whether internal reasoning can meaningfully reduce epistemic uncertainty, or whether external interaction is necessary.

**Training-time Alignment.** After pretraining, a model's internal reasoning capability, reflected in its distribution of internal solvability across tasks, is relatively stable. In contrast, the policy that maps solvability estimates to tool-use decisions remains adjustable during alignment. Training thus plays a critical role in calibrating how agents translate epistemic uncertainty into action. As illustrated in Section 3.4, systematic miscalibration leads to two dominant failure modes. Overestimating internal solvability causes agents to rely on internal reasoning where it is unreliable, resulting in hallucination or incorrect reasoning (Gekhman et al., 2024). Underestimating internal solvability leads agents to invoke external tools unnecessarily, incurring avoidable interaction overhead and inefficiency (Qian et al., 2025). Importantly, these failures arise not from deficiencies in reasoning or tools themselves, but from misaligned meta-cognitive judgments. Effective training therefore aims to calibrate tool-use decisions with respect to internal solvability. Approaches such as supervised fine-tuning with explicit tool-use supervision, or reinforcement learning with task-level feedback, can encourage agents to invoke external tools primarily when internal solvability is low, and to rely on internal reasoning otherwise, as elaborated in Section 4.3.

**Inference-time Alignment.** During inference, agents operate under incomplete and evolving information. Initial contexts may be insufficient to determine whether internal reasoning alone will succeed. By interacting with external tools—such as querying APIs or executing actions—the agent incrementally augments its context, which in turn updates its internal solvability estimate. Inference thus unfolds as a sequential feedback loop, in which the agent alternates between internal reasoning and external interaction while continuously reassessing whether further information is needed. Meta-cognition is essential to regulating this loop: without accurate self-assessment, the agent may terminate prematurely, persist in unproductive reasoning, or overuse tools inefficiently. Robust inference-time behavior emerges when agents can adaptively balance internal reasoning and external interaction in response to changing epistemic conditions.

## 4.2. Behavioral Regimes under Tool-Use Uncertainty

We therefore examine four representative behavior modes defined by how agents allocate epistemic effort between internal reasoning and external interaction to solve the task

successfully, and discuss their consequences for alignment, efficiency, and long-term capability development.

**High internal reasoning and high external tool use.** In this mode, the agent invokes extensive internal reasoning while also frequently calling external tools, regardless of epistemic necessity. Although this behavior may achieve correctness, it is inefficient and obscures the underlying decision logic. The agent neither trusts its internal competence nor relies selectively on external information, effectively treating tool use as brute-force exploration. This pattern reflects poor calibration of internal solvability estimates and leads to unnecessary computational overhead, increased latency, and higher risk of cascading errors.

**Low internal reasoning and high external tool use.** Here, the agent systematically defers to external tools while underutilizing its internal reasoning capability (e.g., from point $B$ to point $A$ for external task; from point $C$ to point $A$ for internal task as shown in Fig 4). This behavior may be effective for weaker models or information-intensive tasks, but it introduces persistent inefficiency and dependence on external systems. More critically, it undermines the goal of model scaling and continual learning: by outsourcing problems that could be solved internally, the agent fails to consolidate and exploit its own parametric knowledge. From the perspective of Section 3, this mode corresponds to underestimating internal solvability.

**High internal reasoning and low external tool use.** In this regime, the agent relies heavily on internal reasoning and avoids external interaction, even when external information could simplify or disambiguate the task (Wang et al., 2025a). This behavior reflects strong internal competence and autonomy, and is desirable in constrained or offline environments. However, excessive internal deliberation can lead to overthinking, long reasoning traces, or brittle inference when required information lies outside the model's internal scope. This mode corresponds to overestimating internal solvability and is a common source of hallucination or logically consistent but factually incorrect outputs.

**Low internal reasoning and low external tool use.** This mode represents the most efficient observable behavior: the agent solves tasks using minimal internal deliberation and invokes external tools only when epistemically necessary (Arora & Zanette, 2025; Wang et al., 2025a) (e.g., point $B$ in Fig 4). Such behavior reflects well-calibrated metacognition and effective alignment between internal solvability estimates and tool-use decisions. However, achieving this regime reliably is non-trivial: overly aggressive minimization risks underthinking or premature termination on complex tasks. Training agents to operate in this regime therefore requires careful balancing of correctness and efficiency signals.

### 4.3. Paths toward Aligned and Efficient Agent Behavior

The behavioral regimes above reveal that alignment is not achieved by maximizing external tool use or maximizing internal reasoning, but by learning when each mode of effort allocation is epistemically justified. This reframes training as a problem of calibrating decisions under uncertainty rather than optimizing a fixed notion of optimal behavior. In the following, we outline several complementary approaches that move agents toward this objective.

**Agentic Midtraining.** Large language models acquire extensive world knowledge through next-token prediction, effectively compressing information into their parametric space (Kaplan et al., 2020). However, this objective alone does not teach agents how to acquire new knowledge through interaction. To address this gap, we advocate extending pretraining with next-tool prediction, where interaction itself becomes a first-class modeling target. By learning to predict which tool to invoke given a context, the agent is trained not only to reason, but to decide how to reduce uncertainty. Modeling interactions (e.g., API calls, UI navigation, or environment actions) as structured outputs enables agents to learn procedural knowledge acquisition, rather than relying solely on static compression. As agent architectures become more unified, this shift opens the door to a new form of scaling: one that governs knowledge acquisition through interaction, rather than knowledge storage alone (Xie et al., 2024b; Li et al., 2024).

**Agentic Supervised Fine-Tuning.** Supervised fine-tuning (SFT) remains a common approach for teaching agents how and when to use tools, often through curated demonstrations on specific tasks (Gou et al.; Li et al., 2025a). However, such approaches frequently assume a uniform internal capability across models. As discussed in Section 3.1, this assumption is unrealistic: what constitutes appropriate tool use for a small model may be redundant or counterproductive for a larger one. One approach is to tailor SFT datasets to each model's internal competence, but this quickly becomes resource-intensive. A more scalable alternative is to train agents to defer selectively when faced with unfamiliar or low-solvability contexts, approximating a conservative upper envelope of internal capability (Qian et al., 2025) (e.g., by assuming some tasks naturally fall into $\mathcal{Q}^{\text{world}} \setminus \mathcal{Q}_{\max}$ in Proposition 3.4). While this improves generality, it may sacrifice precision in specialized domains, illustrating a fundamental trade-off between scalability and fine-grained alignment.

**Agentic Reinforcement Learning: Calibrating Decisions through Experience.** RL provides a natural mechanism

for aligning tool-use decisions with internal solvability, as agents can learn from interaction outcomes rather than static demonstrations. Crucially, effective RL for agents must go beyond correctness-based rewards. Optimizing solely for task success ignores how solutions are reached, allowing inefficient or miscalibrated behaviors to persist (Jin et al., 2025; Li et al., 2025b). Recent methods such as OTC-PO (Wang et al., 2025a) explicitly balance correctness with penalties for unnecessary tool use, encouraging agents to act with restraint. More broadly, RL enables agents to learn process-level preferences: when to reason, when to act, and when to stop. We further envision an iterative training paradigm, RL → SFT → RL, where reinforcement learning discovers aligned trajectories under uncertainty, and supervised fine-tuning consolidates these behaviors for stability and generalization. Such cycles gradually refine both decision policies and meta-cognitive calibration.[5] In addition, it is extremely promising to consider RL during pre-training stage with sufficient computing resources.

**Agentic Prompting.** Prompting-based methods enable agents to exhibit complex tool-use behaviors without parameter updates (Chen et al., 2023; Qiu et al., 2025). While effective in practice, these approaches often lack systematic evaluation of decision quality, allowing overthinking or overacting to persist beneath correct outputs. Recent work incorporating memory or workflow abstractions (Zhang et al., 2024) demonstrates that prompting can guide more efficient behavior, but sustaining alignment typically requires integration with learning-based methods.

**Summary and Implications** Across these approaches, a common theme emerges: improving agent behavior is less about maximizing reasoning or minimizing tool use, and more about calibrating decisions under epistemic uncertainty. Methods that encourage agents to estimate their own internal solvability and allocate effort accordingly—whether through pretraining, supervision, reinforcement learning, or prompting—are better positioned to produce efficient, robust, and adaptable agents. In this sense, alignment is not a fixed target but an emergent property of well-calibrated decision-making. The paths outlined above represent complementary strategies for steering agents away from systematically misaligned regimes and toward behavior that balances reasoning and acting in realistic environments.

## 5. Alternative Views

Intelligent agents have been conceptualized through several dominant paradigms, each emphasizing a different computational bottleneck. We briefly contrast them with our position: 1) *Agents as planners* view decision-making as

plan optimization over an internal world model. This perspective presumes sufficient internal knowledge and offers no principled criterion for deciding when external information must be acquired; 2) *Agents as policy learners* (e.g., in reinforcement learning) treat tool use as just another action optimized for reward. While effective for control, this view does not distinguish epistemically necessary tool use from unnecessary delegation, allowing agents to overuse tools as long as rewards are achieved; 3) *Agents as workflow orchestrators* focus on coordinating tools and procedures to complete tasks reliably. These approaches emphasize execution but largely ignore whether tool use is justified, permitting correctness at the cost of inefficiency and limited internal capability growth.

In contrast, we argue that an agent should invoke external tools only when epistemically necessary. External interaction is justified if and only if the remaining epistemic effort required to complete a task cannot be satisfied through internal reasoning alone. Tool use reallocates epistemic effort; it does not eliminate it. Unnecessary delegation is therefore not merely inefficient—it suppresses the development of internal reasoning capability. This position is most appropriate when the central bottleneck is epistemic efficiency and intelligence development, rather than control optimality or execution speed. It provides a normative criterion for tool use, explains failure modes such as overthinking and overacting, and directly informs training and evaluation. More discussions about long-context, RAG and harness can be found in Appendix B.

## 6. Conclusion

This paper introduces the ***Theory of Agent (ToA)***, a framework that views modern agents as tool-use decision makers operating under epistemic uncertainty. We argue that the central challenge in agent behavior is not how to reason or act, but when external interaction is epistemically necessary. By distinguishing an agent's internal task set from the world task set, ToA formalizes a knowledge boundary that governs rational tool use. Thus, tool use does not eliminate epistemic difficulty but reallocates it. Agents that invoke external tools for internally solvable tasks may achieve correctness yet hinder the development of internal reasoning capability, while agents that rely on internal reasoning beyond their knowledge boundary exhibit overthinking and hallucination. These failure modes stem from misaligned effort allocation rather than deficiencies in reasoning or tools. We hope this framework informs future work on agent alignment, evaluation, and design, contributing to the development of more intelligent agents. We make a blog here [6] and welcome to leave any comments or discussions.

---

[6]https://hrwise-nlp.github.io/assets/websites/theory-of-agent/

---

[5]Additional future directions are discussed in Appendix D.

## Acknowledgement

An initial version of this paper was mainly drafted by Hongru Wang and Cheng Qian during Hongru's visit to UIUC, and was subsequently developed and substantially refined during Hongru's postdoctoral work at the University of Edinburgh. It is also recommended to read the first version at the Arxiv [7]. We sincerely thank the HotDesk group at the University of Edinburgh for their valuable feedback and insightful discussion, with special appreciation to Jushi Kai, Shujia Liu, and Miao Li. We also wish to thank the anonymous reviewers at NeurIPS 2025 and ICML 2026 for their careful reading and exceptionally encouraging comments. We were particularly moved by comments such as "I think this is a very impressive work given the theory-lack nature of computer science & engineering." "The theory here itself is already very good, it does not need to seek references to older, more history-complicated terms." and "The paper takes a genuinely interesting angle on tool use in agents." Such acknowledgments are humbling and motivating in equal measure, and have reaffirmed our commitment to developing a principled theory of agent. Welcome to join us!

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

# A. Explanation about Internal Tools and External Tools

**Internal cognitive tools.** Cognitive tools refer to internal cognitive mechanisms that support systematic or investigative thinking to solve problems (Jonassen, 1992; Kommers et al., 1992). In the context of intelligent agents, various reasoning modules (Zhou et al., 2024; Hongru et al., 2025), such as Chain-of-Thought (Wei et al., 2022), reflection, decomposition (Wang et al., 2025b), and alternative-thinking, function as cognitive processes that enable the retrieval and manipulation of internal knowledge to guide problem-solving. Beyond these, other cognitive tools appear in diverse applications, such as conversational strategies in dialogue systems (Wang et al., 2024a) and psychologically inspired mechanisms designed to model uncertainty, emotion, or user intent (Wang et al., 2023a). Despite their varied forms, these tools share a common function: *they serve as triggers for internal knowledge retrieval, allowing the model to reason and act based on its internal knowledge.*

**External physical tools.** External physical tools refer to modules or interfaces outside the model that are invoked through specific triggers, such as rules, actions, or special tokens, whose outputs are then incorporated into the model's context to inform subsequent reasoning (Hao et al., 2023; Lu et al., 2025). These tools provide access to information or functionality that lies beyond the agent's internal knowledge. Examples include querying a search engine (Jin et al., 2025), calling an API (Wang et al., 2024b), interacting with a user interface (Han et al., 2025), or executing actions in an embodied environment (Li et al., 2024). This perspective enables a unified treatment of diverse forms of interaction as structured tool use: *they serve as interfaces for external knowledge acquisition, allowing the model to access and interact with knowledge beyond its internal capability.*

# B. Discussions

## B.1. Why Over-Delegation Leads to Capability Stagnation

The stagnation effect in Proposition 3.9 is not merely behavioral but arises from the learning dynamics induced by over-delegation. In most training settings, agents are optimized using outcome-based objectives—such as task success, correctness, or sparse terminal reward—without explicit supervision over how uncertainty is resolved. When an external tool is available and reliably produces correct information, invoking it becomes a low-risk strategy that guarantees reward, regardless of whether the task could have been solved internally.

As a result, external tools act as a reward shortcut: they collapse epistemic uncertainty externally before internal reasoning is exercised. This has a direct consequence for learning. Because the agent achieves success without relying on internal reasoning, gradient signals associated with internal cognitive tools become sparse or uninformative. Internal reasoning trajectories are neither required nor reinforced, and therefore receive little to no learning signal. Over time, the policy learns to associate delegation—not reasoning—with reward.

This dynamic closely mirrors reward hacking in reinforcement learning. When the objective only measures task success, the agent naturally exploits the most reliable path to reward, even if that path bypasses the intended capability. External tools "save" the agent from failure, but in doing so, they also shield internal reasoning from both error signals and corrective feedback. The agent becomes increasingly dependent on delegation, while its internal reasoning capability remains under-trained or even degrades relative to what it could have achieved.

Crucially, this stagnation can occur even when internal reasoning is sufficient in principle. The issue is not that the agent lacks capacity, but that the training signal fails to distinguish epistemically necessary tool use from unnecessary delegation. In this sense, over-delegation is not simply inefficient—it actively alters the learning trajectory by suppressing opportunities for internal knowledge consolidation and skill acquisition.

## B.2. Long Context v.s. RAG.

The longstanding debate between long-context modeling and retrieval-augmented generation (RAG) is often framed as a **capacity** question—how much information can fit in the context window, and how reliably the model attends to it. Under the Theory of Agent, this debate acquires a sharper formulation: both approaches *do not eliminate epistemic effort; they only relocate where it is resolved* (Proposition 3.8). Long context is an instance of preemptive context expansion: information that would otherwise require an external retrieval call is injected into the agent's context ahead of time, effectively moving a task from $\mathcal{Q}^{\text{ext}}$ into $\mathcal{Q}^{\text{int}}$ by enlarging the information available for internal reasoning. RAG, in contrast, keeps information external on demand, resolving epistemic uncertainty through targeted tool calls at inference time.

Within ToA, these two strategies are not symmetric. Once a task admits a correct solution through long-context reasoning alone, the agent is operating at or near point $C$ of Fig. 3: pure internal reasoning over an expanded context—whereas repeatedly invoking a retrieval tool for the same task pushes effort allocation toward point $A$, even when the task is by then internally solvable. The normative implication is therefore clear: *conditional on producing a correct answer, long-context reasoning reflects a stronger form of capability consolidation whenever feasible.* This aligns naturally with a prominent direction in frontier labs, where scaling context length and long-context reasoning capability has been pursued as a first-class objective; our framework gives a principled reason for that trajectory—every successful long-context answer corresponds to a task that has been moved into $\mathcal{Q}^{\text{int}}$, supplying the training signal that further consolidates internal capability.

This does not render RAG obsolete. RAG is the epistemically justified strategy whenever context expansion is infeasible or uninformative: when the relevant information exceeds hardware/latency budgets, when it is time-varying (real-time data, live state) and cannot be meaningfully pre-loaded, or when it lies outside any distribution the model has learned to parse. In these cases, external retrieval is not a shortcut but an epistemic necessity, and invoking it is entirely consistent with ToA's normative criterion.

### B.3. Internalization and Externalization.

There is a popular and important viewpoint that ***agents are the combination of model and harness.*** The **model** supplies parametric knowledge and internal reasoning capability, the basis of $\mathcal{Q}^{\text{int}}(m, \mathcal{M})$, while the **harness** supplies tools, memory, context-management routines, and protocols for external interaction, the interface to $\mathcal{Q}^{\text{world}}(\mathcal{M}) \setminus \mathcal{Q}^{\text{int}}(m, \mathcal{M})$. Under the ToA framework, the knowledge boundary is therefore not a property of the model alone; it is co-determined by what the model has learned and what the harness exposes. Importantly, this boundary is not fixed: it is continuously reshaped by two opposing processes: internalization (Lu et al., 2026) and externalization (Zhou et al., 2026).

**Internalization.** Capabilities historically provided through the harness can, under the right training conditions, be absorbed into the model's parametric repertoire, effectively expanding $\mathcal{Q}^{\text{int}}$. Two conditions make a capability a reasonable internalization candidate: (i) the underlying behavior is compressible, it admits structure that can be captured parametrically (e.g., arithmetic, syntactic normalization, certain forms of retrieval) rather than requiring fresh external bits at every invocation; and (ii) the training process supplies a usable learning signal for it, tasks must actually be solved internally at least some of the time, otherwise Prop 3.8 applies and the capability never consolidates. Internalization, in other words, is not automatic: it requires both a compressible target and training conditions that reward internal solutions.

**Externalization.** The reverse direction offloads capabilities to the harness. Externalization is the right choice when information is genuinely time-varying, when actions must produce side-effects in the world, or when external resolution is strictly more reliable or auditable (e.g., verifiers, formal checks). What ToA warns against is externalization by **default**: offloading a capability solely because the current model finds it difficult, rather than because the capability structurally belongs outside the model. Under correctness-only objectives, this default externalization is an attractor (point $A$ or beyond segment $AB$), and it silently freezes the knowledge boundary at the agent's early-training state.

The internalization–externalization split is not a fixed architectural choice; it shifts across training runs and across model generations. Behaviors that today require a harness component may tomorrow sit inside the base model; conversely, new external interfaces (real-time data, embodied actuation, multi-agent protocols) may externalize functions that were previously internal. We do not claim that harnesses will eventually disappear—many external capabilities are **structurally** external and should remain so. We do claim that *the boundary itself is a moving target*, and that its motion is a defining axis of agent development.

This perspective reframes a core design question for future agent systems: not "how large should the model be?" or "how rich should the harness be?" in isolation, but ***how should the internalization–externalization boundary be managed over time***, such that (a) capabilities that can be internalized are given training conditions to be so, (b) capabilities that must remain external are not mistaken for candidates for internalization, and (c) the split between $\mathcal{Q}^{\text{int}}$ and $\mathcal{Q}^{\text{ext}}$ continues to expand in the direction of greater autonomy. We see this as an open and important research direction, and one that ToA is well-positioned to frame: it supplies the language for making this moving boundary explicit, and future work should make it trainable.

### B.4. Self-evolving Agents

An agent is *self-evolving* when its internal task set strictly expands over time.

$$\mathcal{Q}^{\text{int}}(m_t, \mathcal{W}) \ \subseteq \ \mathcal{Q}^{\text{int}}(m_{t+1}, \mathcal{W}), \tag{12}$$

where $m_t$ denotes the agent at time step $t$. Whether this expansion is *sufficient* depends on whether the surrounding world is itself moving.

**Static world.** When the world task set $\mathcal{Q}^{\text{world}}(\mathcal{W})$ is fixed, self-evolution is a *coverage* problem: the knowledge boundary drifts outward, reclaiming tasks that previously required external delegation. The asymptotic goal is

$$\mathcal{Q}^{\text{int}}(m_t, \mathcal{W}) \ \xrightarrow{t \to \infty} \ \mathcal{Q}^{\text{world}}(\mathcal{W}), \tag{13}$$

i.e. eventually everything the world affords can be solved internally.

**Dynamic world.** Real worlds emit new tasks continually: the environment $\mathcal{W}_t$ itself grows with $t$. Evolution then becomes a *rate* condition — the agent must internalize tasks at least as fast as the world produces them:

$$\frac{d}{dt}\left|\mathcal{Q}^{\text{int}}(m_t, \mathcal{W}_t)\right| \ \geq \ \frac{d}{dt}\left|\mathcal{Q}^{\text{world}}(\mathcal{W}_t)\right|. \tag{14}$$

If inequality Eq. 14 fails, the knowledge boundary stagnates *relative* to the world, and external delegation remains structurally necessary no matter how much $m_t$ improves in isolation. Self-evolution in a static world is about *closing the gap*; in a dynamic world, it is about *not falling behind*.

## C. Impact Statements

This paper advances a conceptual understanding of tool-augmented agents by arguing that tool-use decisions shape not only task efficiency but the long-term development of agent intelligence. By introducing epistemic necessity as a normative criterion for external interaction, the Theory of Agent (ToA) reframes tool use from a performance optimization problem to a learning-critical decision under uncertainty. This perspective has practical implications for the design, training, and evaluation of autonomous agents: agents that over-delegate to external tools may achieve short-term correctness but risk stagnating their internal reasoning capability, while epistemically calibrated agents can preserve and expand internal competence over time.

Beyond improving efficiency or reducing hallucinations, this work highlights a broader societal implication: as agents increasingly rely on external systems, unprincipled delegation may entrench dependency, fragility, and limited autonomy. By emphasizing when external interaction is epistemically necessary rather than merely convenient, our framework encourages the development of agents that learn more effectively, generalize better, and remain robust in settings where external access is constrained or unreliable. We hope this work informs future research on agent alignment, evaluation, and long-horizon learning, contributing to the development of more capable and autonomous machine intelligence.

## D. Future Directions

**Vision Agent.** Vision agents extend our unified framework of reasoning and acting by incorporating visual affordances as part of the decision-making loop. In our definition, external physical tools are invoked based on an agent's knowledge gaps; in vision agents, visual input becomes a direct means of detecting such gaps and informing tool use decisions. To realize this, future systems should treat visual understanding not as passive recognition but as actionable epistemic input. This involves embedding affordance-aware modules into vision-language models that not only recognize objects but predict possible interactions. Moreover, meta-cognitive control should guide visual attention: the agent must actively attend to regions most likely to resolve its uncertainty. Training in simulation with reinforcement learning can allow agents to learn the utility of visual exploration for acquiring external knowledge, enabling more precise tool invocation grounded in perception.

**Embodied Agent.** Embodied agents concretize the external physical tool dimension by extending it into the physical world, where the agent's own body becomes a tool, and the environment imposes dynamic constraints. Within our framework,

this embodiment means that the agent's knowledge boundary is not only cognitive but also physically bounded (e.g., what can be seen, reached, or manipulated). To operationalize this, agents should be equipped with real-time sensorimotor feedback loops and control modules that treat actions as epistemic moves: physical actions (e.g., MoveTo, PickUp) should be treated like external tool calls that yield knowledge from the environment. Learning here must be closed-loop and incremental—using reinforcement signals from physical interaction to adjust the decision boundary over time. Physical meta-cognition, such as failure detection or confidence in execution, should guide whether to reason further, retry an action, or explore alternatives.

**Multi-Agent Coordination.** Multi-agent coordination extends our framework from individual agents aligning their decision and knowledge boundaries to a collective setting where these boundaries are distributed across multiple agents. In this paradigm, each agent operates with a local view (its own knowledge and decision boundaries), but contributes to a shared task by reasoning about and interacting with other agents. The key challenge is aligning these distributed boundaries to form a coherent collective intelligence. To achieve this, agents must be equipped with mechanisms to communicate epistemic state, and dynamically delegate subtasks to peers whose knowledge boundaries better match the problem context. This requires structured communication protocols, role inference strategies, and shared meta-cognitive modules that manage when to ask, respond, or act. Practically, this can be developed through multi-agent reinforcement learning in environments where cooperation is required for successful task completion, with reward functions encouraging efficient division of cognitive and physical labor.

