# OpenReview forum: "Position: Agents Should Invoke External Tools ONLY When Epistemically Necessary"
_ICML.cc/2026/Position_Paper_Track — ICML 2026 Position Paper Track regular_

### Official Review · Reviewer_W3f4 · 2026-02-16

**Significance:** 3
**Argument Clarity:** 2
**Rating:** 4
**Confidence:** 3

**Questions:**

1. Given that the internal task set is defined as model-specific and latent, how do the authors propose to objectively benchmark the epistemic necessity of a tool call in a way that is reproducible across different architectures?

2. In the proposed RL→SFT→RL cycle, how can we ensure that the penalty for unnecessary tool use does not inadvertently encourage hallucination when the model is at its knowledge boundary?

3. Does the theory account for the varying computational "cost" of internal reasoning (e.g., long Chain-of-Thought) versus the cost of external calls, and can these costs be unified within the epistemic effort framework?

**Alternative Views Section:**

Yes

**Compliance With Llm Reviewing Policy A Conservative:**

Affirmed.

**Discussion Potential:**

3

**Paper Summary:**

The paper presents the Theory of Agent (ToA), a framework that reframes the interaction between an agent's internal reasoning and its use of external tools. The authors advocate for a normative principle where agents should invoke external tools only when it is epistemically necessary, meaning the task cannot be reliably resolved through internal reasoning alone. By defining internal and world task sets, the paper identifies a latent knowledge boundary that governs these decisions. A core contribution is the concept of epistemic effort as a task-invariant requirement that can be reallocated but not eliminated. The authors argue that unnecessary delegation leads to capability stagnation because the model bypasses opportunities to consolidate internal knowledge. The work concludes by suggesting that agent alignment should be evaluated based on effort-consistent decision-making rather than simple task success.

**Position:**

Yes

**Position In Title:**

Yes

**Related Work:**

3

**Strengths And Weaknesses:**

The paper addresses a highly relevant and timely topic for the ICML community, as the deployment of tool-augmented LLMs often ignores the long-term cognitive costs of over-reliance on external APIs. One major strength is the formalization of the knowledge boundary and the introduction of the epistemic effort invariant, which provides a principled way to analyze common failure modes like overthinking and overacting. The conceptual link between over-delegation and the suppression of gradient signals for internal reasoning is a compelling theoretical argument for why current agents might hit a performance ceiling. The paper is clearly argued and provides a useful taxonomy of behavioral regimes based on effort allocation. However, a significant weakness is the reliance on latent variables like the internal task set and minimal required effort, which are difficult to measure empirically. While the authors propose next-tool prediction and calibrated reinforcement learning as pathways, the paper lacks concrete experimental data or a proposed benchmark to validate that these training methods actually move the knowledge boundary. Additionally, while it cites foundational work like ReAct and LeCun’s autonomous intelligence vision, it could more deeply explore the trade-offs in specialized domains where the cost of internal reasoning (latency) might outweigh the epistemic benefits of autonomy.

**Support:**

2

---

> ### Author Rebuttal · Authors · 2026-03-27
>
> We appreciate the recognition of the knowledge boundary formalization and gradient-signal suppression argument as compelling contributions, and the "good" significance and discussion potential ratings. We address each concern and question directly.
>
> `W1`: Latent ≠ unmeasurable — same as the value function in RL or the true data distribution in generative modeling. Latency motivates principled approximation, not abandonment. Concretely: $p^{int}_t$ is approximated by (a) *self-consistency rollout* — K internal traces without tool access; high output agreement = high solvability — or (b) *draft-reasoning confidence* — single forward pass using output probability as threshold signal. (c) *hidden-state probes*: a linear probe on intermediate activations estimates $p^{int}_t$ at near-zero overhead, enabling inference-time routing without full rollout. E* is approximated by the minimum tokens+tool-calls across correct trajectories, as in OTC-PO [1], which tracks the minimum correct-trajectory tool count across rollouts as a practical E*_ext surrogate. OSWorld-Human [2] human-annotated optimal trajectories provide a reference E* for computer-use tasks. We will add an operational instantiation subsection connecting these proxies to the RL→SFT→RL cycle in §4.3.
>
> `W2`: As a position paper we focus on principled arguments; there are two existing lines of benchmarks directly operationalize our evaluation proposals. (1) *Efficient reasoning* [3]: benchmarks token efficiency across 5 math datasets; RL length penalties reduce unnecessary tokens 65% on GSM8K with <2% accuracy loss — demonstrating E_int misallocation is measurable and correctable across architectures. (2) *Efficient acting* via OSWorld-Human [2]: 369 tasks with human-annotated optimal trajectories across 9 applications; 16 agents evaluated, all showing 1.4–2.7× step overhead. OTC-PO [1] reduces tool calls 68.3% and improves tool productivity 215.4% while maintaining accuracy — validating "correct but overacting" (Corollary 3.10) across multiple architectures. (3) SMART [4] directly validates the Q_int/Q_ext distinction is learnable: metacognitive step annotations reduce tool overuse 24% and improve performance 37%.
>
> `W3`: The effort decomposition extends to graded costs: $E_{total} = c_{int} · E_{int} + c_{ext} · E_{ext}$ where $c_{int}$ captures per-token cost (quadratic attention, KV cache growth) and $c_{ext}$ captures per-call cost (API latency, monetary fees). When CoT cost is high relative to external call cost, rational delegation shifts the ideal operating point toward A in Figure 3 — justified by resource rationality even without strict epistemic necessity. α implicitly encodes $c_{int}/c_{ext}$: high internal cost → lower α is rational. The slope in Figure3 exactly stands for this ratio and this connects to resource-bounded rationality; we will incorporate this near Eq. 4.
>
> `Q1`: See W2 above. OSWorld-Human [2] provides architecture-agnostic benchmarking via human-annotated optimal trajectories; tool productivity from OTC-PO [1] is a cross-architecture overacting metric (correct answers per tool invocation). For overthinking, Arora & Zanette [3] provide length-accuracy tradeoff curves across model sizes and datasets.
>
> `Q2`: RLVR outcome-based rewards address hallucination directly: incorrect internal answers receive zero reward and are penalized — wrong hallucinations are immediately suppressed. If a hallucinated answer happens to be correct (overthinking failure mode), it is addressable by RL efficiency rewards penalizing unnecessary reasoning tokens [3], incentivizing correct but concise chains rather than verbose-but-lucky ones. The SFT phase provides structural reinforcement via process supervision — training on step-verified reasoning teaches accurate internal chains. The RL→SFT→RL cycle provides complementary pressures: SFT teaches *how* to reason accurately; RL teaches *when* to stop and delegate. OTC-PO also counters Prop 3.8 stagnation: penalizing excess tool calls among correct trajectories forces the model to exercise internal reasoning, preventing delegation-as-default.
>
> `Q3`: Addressed in W3 above. The graded-cost formulation unifies CoT and external call costs within the epistemic effort framework. Geometrically, varying $c_{int}/c_{ext}$ shifts the ideal operating point along segment AC in Figure 3: high ratio → closer to A (prefer external); low ratio → closer to C (prefer internal). α naturally encodes this ratio — deployments with expensive internal compute warrant lower α; those with costly API calls warrant higher α. This makes the CoT-vs-external-call tradeoff an explicit, principled design parameter.
>
> [1] Teaching Language Model to Act Efficiently, NeurIPS LAW Workshop 2025
>
> [2] OSWorld-Human: Benchmarking the Efficiency of Computer-Use Agents, ICML Workshop on Computer-Use Agents 2025
>
> [3] Training Language Models to Reason Efficiently, NeurIPS 2025
>
> [4] SMART: Self-Aware Agent for Tool Overuse Mitigation, ACL Findings 2025

---

### Official Review · Reviewer_JCD9 · 2026-03-12

**Significance:** 4
**Argument Clarity:** 2
**Rating:** 4
**Confidence:** 4

**Questions:**

How should practitioners estimate the operational quantity pint_t(q,m;M) in real time? What signals (e.g., confidence scores, internal debate patterns, intermediate verification steps) should serve as proxies for true internal solvability?

Proposition 3.8 (line 224) relies on outcome-based objectives without explicit tool-use supervision. How robust is this effect to different training regimes — if agents are trained with explicit meta-cognitive loss terms that penalize unnecessary delegation, does the stagnation effect persist?

**Alternative Views Section:**

Yes

**Compliance With Llm Reviewing Policy A Conservative:**

Affirmed.

**Discussion Potential:**

3

**Final Justification:**

I think the paper's position is very clear and reasonable to me. I like the formulation of overthinking and overacting, which are very good clarification for the failure modes for current agents. Overall, I lean to accept this paper.

**Paper Summary:**

This position paper introduces the Theory of Agent (ToA), a framework for understanding **when tool-augmented agents should invoke external tools**. The core thesis is that agents should invoke external tools only when epistemically necessary. If they are remaining uncertainty required for task completion, and this cannot be resolved through internal reasoning alone, they should call the tools.

The authors formalize this position by distinguishing between internal task sets (tasks solvable via reasoning alone) and world task sets (all tasks solvable with external interaction), introducing the concept of a knowledge boundary separating them. They define epistemic effort as an invariant requirement that cannot be eliminated, only reallocated between internal reasoning and external interaction.

The paper argues that **unnecessary delegation not only introduces inefficiency but actively impedes the development of internal reasoning capability**. Common failure modes like overthinking and overacting are reframed as misallocations of effort rather than deficiencies in reasoning or tools.

The authors discuss implications for training (meta-cognition calibration), inference-time behavior, and agent design, proposing approaches including agentic pretraining with next-tool prediction, supervised fine-tuning, reinforcement learning, and prompting methods that encourage better effort allocation.

**Position:**

Yes

**Position In Title:**

Yes

**Related Work:**

3

**Strengths And Weaknesses:**

## Strengths:

- The paper articulates a well-defined position that agents should invoke tools only when epistemically necessary, providing a criterion rather than relying on reward optimization. The formal framework is rigorous and the insight is valuable.

- The treatment of reasoning and acting as alternative tools for knowledge acquisition is appealing, and the four behavioral regimes in Section 4.2 effectively illustrate how different effort allocations lead to distinct failure modes.

- The paper engages seriously with practical implementation paths (agentic pretraining, SFT, RL, prompting) and provides a well-motivated critique of existing paradigms (agent-as-planner, agent-as-policy-learner, workflow-orchestrator).

## Weaknesses

- The paper might want to provide more empirical evidence that explains observed agent failures or that agents trained under these principles develop better long-term reasoning capabilities.

- The gap between effort allocation theory and actual training objectives is significant. The paper discusses training approaches but does not provide a principled way to construct objectives that directly optimize without reward hacking.

**Support:**

3

---

> ### Author Rebuttal · Authors · 2026-03-27
>
> We appreciate the reviewer's recognition of the well-defined position, the appealing treatment of reasoning and acting as alternative knowledge-acquisition tools, the effective illustration of four behavioral regimes (Section 4.2), and the practical engagement with implementation paths. Regarding weakness and questions:
>
> `W1 Needs more empirical evidence`
>
> We point to the following converging lines of evidence that directly instantiate ToA's core claims as listed in paper:
>
> **Overthinking**: Cuadron et al. [1] and Arora & Zanette [2] document empirically that models over-allocate internal computation on tasks they could answer more concisely — a direct manifestation of Corollary 3.10's overthinking failure mode. Arora & Zanette demonstrate that RL-based length penalties can correct this without accuracy loss, with better effort allocation.
>
> **Overacting**: SMART [3] shows that LLMs invoke external tools unnecessarily over 30% of the time on internally-solvable tasks, and that tool overuse can degrade performance. OSWorld-Human [4] shows that even the strongest computer-use agents take 1.4–2.7× more steps than human-optimal trajectories, with the best agent achieving only 17.4% on the strictest efficiency metric despite 42.5% task success. OTC-PO [5] reduces tool calls by up to 68.3% and improves tool productivity by up to 215.4%, while maintaining comparable accuracy — directly validating the "correct but overacting" diagnosis of Corollary 3.10. Moreover, OTC-PO's result that models trained with efficiency signals maintain internal reasoning quality provides direct support for the contrapositive of Proposition 3.8.
>
> We will expand the discussion of this evidence in the main text and connect each major claim to specific numbers from these works.
>
> `W2 Gap between effort allocation theory and training objectives; reward hacking risk`
>
> We provide several actionable strategies and training objectives at Section 4. For examples, there are many paper listed to address the overthinking and overacting issues. One of them, OTC-PO [5] provides the most direct bridge: its **correctness-gated efficiency reward** is a concrete instantiation of the ToA alignment criterion. Formally, the OTC-PO reward function is:  $r(q, m) = r_{efficiency(q, m)} · 𝟙[r_{correct(q, m)} > 0]$. The gating on correctness ensures that efficiency optimization only activates on already-correct trajectories, directly alleviating the reward hacking concern: the model cannot "hack" the efficiency reward by sacrificing correctness. There are several other studies that operationalizes effort-consistent alignment within standard RL pipelines, and we will provide more detailed references in next version.
>
> `Q1`: We acknowledge that $p^{int}_t(q,m;M)$ and $E^*(q,m)$ are latent. However, the *latency* of these quantities does not make them unoperationalizable — it means they must be estimated. Concretely:
>
> **Some of proxy signals for $p^{int}_t(q,m;M)$** (internal solvability at step $t$):
> - **Self-consistency sampling**: sample K independent reasoning traces without tool access; high agreement (k/K > threshold) estimates high internal solvability.
> - **Confidence probes on hidden states**: lightweight classifiers trained on intermediate layer activations to predict task internal solvability.
>
> The key operational insight is: **$p^{int}_t$ can be approximated by practical rollout at inference time** — the agent samples or drafts an internal answer and uses its confidence in that draft to decide whether to delegate. This does not require knowing the ground-truth knowledge boundary; it only requires the agent to have a calibrated sense of its own output quality (a.k.a., self-awareness), which is precisely what SFT and RL alignment trains. In addition, it can also guide the evaluation and diagnosis of existing agents as mentioned in previous answer for the overthinking and overacting issues.
>
> `Q2`: If the training regime includes explicit meta-cognitive supervision (e.g., OTC-PO's correctness-gated efficiency reward, SMART's knowledge-boundary-annotated SFT), Proposition 3.8's stagnation effect is directly mitigated. This is in fact the *design goal* of OTC-PO: by explicitly rewarding efficient tool use among correct trajectories, the training signal ensures that internal reasoning is exercised on internally-solvable tasks, preserving the gradient flow to internal cognitive tools. OTC-PO's empirical results (maintained or improved internal reasoning quality despite reduced tool calls) confirm this prediction.
>
>
> [1] The Danger of Overthinking: Examining the Reasoning-Action Dilemma in Agentic Tasks: https://arxiv.org/abs/2502.08235
>
> [2] Training Language Models to Reason Efficiently, NeurIPS 2025
>
> [3] SMART: Self-Aware Agent for Tool Overuse Mitigation, ACL 2025
>
> [4] OSWorld-Human: Benchmarking the Efficiency of Computer-Use Agents
>
> [5] Acting Less is Reasoning More! Teaching Model to Act Efficiently, LAW Workshop at NeurIPS 2025

---

> > ### Author Rebuttal · Reviewer_JCD9 · 2026-04-03
> >
> > Thanks for the response. I have checked the new mentioned papers.

---

### Official Review · Reviewer_X9Bb · 2026-03-13

**Significance:** 3
**Argument Clarity:** 3
**Rating:** 5
**Confidence:** 5

**Questions:**

&nbsp;

1. In the authors' categorization of reasoning and external tool usage as methods to reduce uncertainty did the authors consider demarcating between aleatoric and epistemic uncertainty? For example what would be the argument against describing resaoning as a measurement apparatus that seeks to reduce the aleatoric noise over an observation?

2. In Definition 3.6, why can't the confidence threshold take on values 0 or 1 i.e. why is $\alpha$ contained within an open interval?

3. Line 352-364 the authors introduce the idea that agents may fail to learn from using external tools in place of internal reasoning. How does this mechanism actually manifest? Can the authors give an example where a model would update its weights as a result of using internal reasoning in place of external tools? Is the authors' intention that e.g., expert iteration approaches or model distillation would be undertaken by SFTing on reasoning traces that leveraged internal reasoning?

4. In Appendix C the authors state, "This has a direct consequence for learning. Because the agent achieves success without relying on internal reasoning, gradient signals associated with internal cognitive tools become sparse or uninformative.". What gradient-based learning methods did the authors have in mind for language agents?

&nbsp;

**Alternative Views Section:**

Yes

**Compliance With Llm Reviewing Policy A Conservative:**

Affirmed.

**Discussion Potential:**

3

**Ethical Review Concerns:**

&nbsp;

No ethical concerns identified.

&nbsp;

**Final Justification:**

&nbsp;

The main concern raised in my initial review was the redefinition of a language agent which I viewed as superfluous in light of existing frameworks. Through the rebuttal phase the authors responded promptly and in detail regarding their proposed changes. As such, I am happy to recommend acceptance for the paper.

&nbsp;

**Paper Summary:**

&nbsp;

The authors propose Theory of Agent (ToA) in support of their position that agents should invoke external tools only when internal reasoning is insufficient to solve a problem. While the position is interesting, my primary concern is whether a redefinition of a language agent is necessary to support the authors' position. In particular, I would like to see more discussion of related work on language agent definitions and why the authors' work is not compatibible with those definitions. I will be inclined to increase my score if the authors can adequately address this concern in the rebuttal phase.

&nbsp;

**Position:**

Yes

**Position In Title:**

Yes

**Related Work:**

3

**Strengths And Weaknesses:**

&nbsp;

The main strength of the paper is that the authors' point is clearly valid. Agents (however they may be defined) should benefit from knowledge of when to apply an external tool and when to apply internal reasoning to solve a problem.

&nbsp;

Below, I demarcate my perceived weaknesses into major and minor points.

&nbsp;

**__MAJOR POINTS__**

&nbsp;

1. The redefinition of an agent posited in the paper conflicts with several prior works e.g. POMDP-based definitions [5, 6]. Furthermore, neither internal nor external tools are necessary (depending on the definition of a "tool") for a language agent to learn from interaction with an environment. The authors' definition heavily implies that BOTH internal or external tools are required to define an agent. To avoid over-proliferation of agent definitions in the literature it would be great if the authors could seriously consider whether there intended definition fits within the scope of an existing definition. Other literature that should be discussed includes the work of [7].

&nbsp;

**__MINOR POINTS__**

&nbsp;

1. There are some missing capitalizations in the references e.g., "ReAct", "LLMs", and "AI".

2. There are some missing confernce citations in the references e.g., [1] was accepted at NeurIPS 2025, [2] was accepted at IJCAI 2024, [3] was accepted at EMNLP 2024, and [4] was published at EMNLP 2025.

3. The choice of terminology, "Theory of Agent" (ToA) is poor in my view since it suggests a broader scope than agentic tool use.

4. In Definition 3.1 the relation between a task and the environment is not clear.

5. In Definition 3.3 the notation of $\mathcal{M}_{agents}$ conflicts with the usage of $\mathcal{M}$ for the environment.

&nbsp;

**__REFERENCES__**

&nbsp;

[1] Arora, D. and Zanette, A., [Training Language Models to Reason Efficiently](https://openreview.net/forum?id=AiZxn84Wdo). In The Thirty-ninth Annual Conference on Neural Information Processing Systems. 2025.

[2] Chen, G., Dong, S., Shu, Y., Zhang, G., Sesay, J., Karlsson, B., Fu, J. and Shi, Y., 2024, August. [AutoAgents: a framework for automatic agent generation](https://dl.acm.org/doi/abs/10.24963/ijcai.2024/3). In Proceedings of the Thirty-Third International Joint Conference on Artificial Intelligence (pp. 22-30).

[3] Gekhman, Z., Yona, G., Aharoni, R., Eyal, M., Feder, A., Reichart, R. and Herzig, J., 2024, November. [Does fine-tuning LLMs on new knowledge encourage hallucinations?](https://aclanthology.org/2024.emnlp-main.444/). In Proceedings of the 2024 Conference on Empirical Methods in Natural Language Processing (pp. 7765-7784).

[4] Li, C., Xue, M., Zhang, Z., Yang, J., Zhang, B., Yu, B., Hui, B., Lin, J., Wang, X. and Liu, D., 2025, November. [Start: Self-taught reasoner with tools](https://aclanthology.org/2025.emnlp-main.683/). In Proceedings of the 2025 Conference on Empirical Methods in Natural Language Processing (pp. 13523-13564).

[5] Christianos, F., Papoudakis, G., Zimmer, M., Coste, T., Wu, Z., Chen, J., Khandelwal, K., Doran, J., Feng, X., Liu, J. and Xiong, Z., 2023. [Pangu-agent: A fine-tunable generalist agent with structured reasoning](https://arxiv.org/abs/2312.14878). arXiv preprint arXiv:2312.14878.

[6] Narayanan, S., Braza, J.D., Griffiths, R.R., Ponnapati, M., Bou, A., Laurent, J., Kabeli, O., Wellawatte, G., Cox, S., Rodriques, S.G. and White, A.D., 2024. [Aviary: training language agents on challenging scientific tasks](https://arxiv.org/abs/2412.21154). arXiv preprint arXiv:2412.21154.

[7] Sumers, T., Yao, S., Narasimhan, K.R. and Griffiths, T.L., 2023. [Cognitive architectures for language agents](https://openreview.net/forum?id=1i6ZCvflQJ). Transactions on Machine Learning Research.

&nbsp;

**Support:**

3

---

> ### Author Rebuttal · Authors · 2026-03-27
>
> We appreciate the recognition that our position is "clearly valid" with good significance and discussion potential, and the explicit willingness to raise the score if the agent definition concern is addressed.
>
> `Major Point 1`: **ToA does not conflict with POMDP-based formalisms** — it introduces a normative criterion orthogonal to them. POMDP agents (Pangu-agent, Aviary) answer *what an agent is* and *how it acts*: a policy mapping observations to actions over a state space. ToA answers a different question: *when, among all valid actions, should external interaction be preferred over internal processing?* This is precisely what POMDP reward functions leave unspecified — they capture task success but say nothing about the provenance of the decision. The tool partition $T=T_{int} \cup T_{ext}$ (§2.2, line 141) maps directly onto any POMDP action space; ToA then adds a normative layer on top via Definition 3.6 and Eq. 7, specifying when each class should be preferred. The underlying state space, transition dynamics, and policy structure are entirely untouched. ToA is thus a normative overlay, fully compatible with existing formalisms.
>
> ToA is also a **general extension of ReAct** (§2.2): ReAct is a special case where internal reasoning steps are monolithic thought units and external actions are tool calls, without any principled epistemic routing criterion. ToA subsumes this by making the routing decision explicit and principled. On CoALA (Sumers et al. 2023): this is a complementary architectural framework. Within ToA, short-term memory fits into interaction context τ_t (Def 3.5); long-term retrieval-based memory is naturally modeled as a form of external tool use. We will add this discussion in revision.
>
> `Minor`: We will fix all capitalization errors, add all recommended references, rename $M_{agents}$ to resolve notation collision with environment M, clarify that Q is determined by the environment, and consider renaming ToA to EToA, which E stands for epistemic.
>
> `Q1 Aleatoric vs. epistemic uncertainty`: Epistemic uncertainty — arising from gaps in the agent's internal knowledge (e.g., not knowing a historical date or fact beyond training cutoff) — is ToA's primary subject; tool use is appropriate when this gap cannot be closed via internal reasoning (p^int_t < α per Def 3.6). Aleatoric uncertainty — irreducible environmental randomness (e.g., current weather, real-time prices) — cannot be resolved internally under any circumstance; tool use is always epistemically justified. Figure 3(b) captures this geometrically: for Q_ext tasks, no pure-internal path (Point C) achieves task success; minimum external effort is structurally required and cannot be traded away. This contrasts with Figure 3(a) for Q_int tasks, where Point C remains a feasible allocation on the solvable frontier.
>
> `Q2 Why α∈(0,1)?`: α=0 makes every delegation justified, reducing ToA to an unconstrained agent with no normative criterion; α=1 requires perfect internal certainty before any internal resolution — practically impossible given calibration imperfections in real models. Neural language models compute solvability estimates via softmax distributions over output tokens, which map strictly to (0,1) with any finite set of logits. A model's confidence in any particular answer never exactly reaches 0 or 1. The open interval α∈(0,1) therefore reflects the computational reality of real neural models, not merely a mathematical convention.
>
> `Q3+Q4 Stagnation mechanism and gradient methods`: In outcome-based RL (GRPO/PPO), external tools that reliably produce correct answers become low-risk reward shortcuts; gradient signals for internal cognitive tools become sparse — the policy learns that delegation, not internal reasoning, is the reliable path to reward. This closely parallels two empirical research directions: (1) *Overthinking* [1,2]: RL length penalties correct over-allocation of internal tokens without accuracy loss — showing the same RL principle applies symmetrically to both E_int and E_ext misallocations. (2) *Overacting* [3,4]: SMART shows >30% unnecessary tool calls on internally-solvable tasks; OTC-PO's correctness-gated efficiency reward reduces tool calls 68.3%. Both SFT (SMART [3]) and RL (OTC-PO [4]) route the gradient back to internal reasoning; stagnation is not inevitable when training rewards epistemic routing.
>
> [1] Cuadron et al. The Danger of Overthinking: Examining the Reasoning-Action Dilemma in Agentic Tasks, 2025
>
> [2] Arora & Zanette. Training Language Models to Reason Efficiently, NeurIPS 2025
>
> [3] Qian et al. SMART: Self-Aware Agent for Tool Overuse Mitigation, ACL Findings 2025
>
> [4] Wang et al. Teaching Language Model to Act Efficiently, NeurIPS LAW Workshop 2025

---

> > ### Author Rebuttal · Reviewer_X9Bb · 2026-04-02
> >
> > &nbsp;
> >
> > Many thanks to the authors for their rebuttal. All minor points are addressed in the authors' response. However, I still have issues with the authors' redefinition of a language agent as in Section 2.2.
> >
> > To be clear, in my initial review I was trying to convey the point that there is no explicit need to redefine a (language) agent as an entity with access to internal and external tools. Indeed, a language agent may be defined without access to any tools as in e.g., the SayCan framework [1]. Rather than redefine a language agent completely and claim this as a contribution of the paper, I would rather the authors operated within an existing and more general POMDP framework e.g., the framework introduced in Aviary presents a more general definition of a language agent which does not necessitate the subdivision of tools into internal and external sets.
> >
> > The authors state that, "ToA is thus a normative overlay, fully compatible with existing formalisms.". I would simply ask that the paper is written in such a way as to make this clear e.g. by avoiding the redefinition of a language agent in Section 2.2 and instead opting to introduce ToA on top of an existing general POMDP framework.
> >
> > &nbsp;
> >
> > **__REFERENCES__**
> >
> > &nbsp;
> >
> > [1] Brohan, A., Chebotar, Y., Finn, C., Hausman, K., Herzog, A., Ho, D., Ibarz, J., Irpan, A., Jang, E., Julian, R. and Kalashnikov, D., 2023, March. [Do as I can, not as I say: Grounding language in robotic affordances.](https://proceedings.mlr.press/v205/ichter23a) In Conference on robot learning (pp. 287-318). PMLR.
> >
> > &nbsp;

---

### Official Review · Reviewer_Vhxo · 2026-03-13

**Significance:** 2
**Argument Clarity:** 3
**Rating:** 4
**Confidence:** 4

**Questions:**

1.The paper models internal reasoning operations as “internal tools.” Could the authors further justify why this abstraction is theoretically appropriate, and clarify what is gained by treating internal reasoning in the same tool-based framework as external actions?

2.The paper uses a threshold α to translate internal solvability estimates into tool-use decisions. Since the knowledge boundary is described as latent, context-dependent, and dynamically shifting, is a thresholded decision rule too simplified to capture this boundary in a principled way?

3.The effort-allocation principle is central to the paper, but it currently appears more conceptual than operational. Could the authors explain more concretely how internal effort and external effort could be instantiated or measured in practice, and in what sense Proposition 3.7 provides a strong principled foundation if these quantities remain difficult to quantify?

4.The paper draws a strong normative distinction between internal reasoning and external tool use, but many practical cases are more hybrid. For example, a task may be internally solvable yet still better handled with a tool because of efficiency, precision, safety, or reliability. In addition, tool outputs may help verify or calibrate internal knowledge. Under such scenarios, should “over-allocation” to external tools still be considered a mistake?

5.What concrete empirical evidence would the authors regard as the strongest validation of Proposition 3.8 on delegation-induced capability stagnation? For example, would this require a longitudinal comparison between agents encouraged to over-delegate and agents trained to solve internally whenever possible?

6.How should the proposed position be adapted for safety-critical or high-stakes settings, where external checking may be desirable even when internal reasoning appears sufficient? Would repeated verification still count as unnecessary delegation in such cases?

7.The paper argues that correctness with excessive tool use may still reflect misalignment. What would a benchmark or evaluation protocol look like that can reliably distinguish a merely correct agent from an epistemically well-calibrated one?

8.The framework emphasizes long-term internal capability development, but many practical deployments prioritize latency, bounded compute, and reliability. How do the authors suggest balancing their normative objective against these competing system goals?

9.The paper discusses future training directions such as reinforcement learning and next-tool prediction. Could the authors clarify more concretely how these proposed objectives differ from existing action prediction or policy learning setups, and what distinct agent behaviors they are expected to produce?

**Alternative Views Section:**

Yes

**Compliance With Llm Reviewing Policy A Conservative:**

Affirmed.

**Discussion Potential:**

3

**Ethics Review Area:**

["Other Expertise"]

**Final Justification:**

Thanks for your detailed response. Overall, the current recommendation is appropriate.

**Paper Summary:**

This paper presents a position on tool-augmented agents: external tools should be invoked only when they are epistemically necessary, that is, when the remaining uncertainty of a task cannot be resolved reliably through internal reasoning over the current context. Rather than treating reasoning and acting as separate pipeline stages, the paper reframes them as two alternative means of knowledge acquisition and formalizes this view through the Theory of Agent. The framework introduces the notions of an internal task set, a world task set, a latent knowledge boundary, belief-based solvability estimates, and epistemic effort as the requirement that must be allocated between internal reasoning and external interaction. Based on this perspective, the paper explains failure modes such as overthinking and overacting, and discusses implications for evaluation, training, and long-term capability development of agents.

**Position:**

Yes

**Position In Title:**

Yes

**Related Work:**

3

**Strengths And Weaknesses:**

Strengths

1.The paper takes a genuinely interesting angle on tool use in agents. A major strength is that the paper does not view the problem merely as selecting among external tools. Instead, it argues that the agent is already making a decision at an earlier stage: whether the remaining uncertainty should be handled through internal reasoning or external interaction. This reframing is conceptually fresh and helps elevate the discussion from “which tool to use” to “whether tool use is justified at all”.

2.The topic is highly relevant to the current agent research landscape and well aligned with ICML community interests. Tool-augmented LLM agents, reasoning efficiency, and agent training are all highly active topics. The paper addresses a frontier question in this area: how to reason about the necessity, rather than just the utility, of tool invocation. This makes the submission timely and likely to attract attention from readers interested in agents, reasoning models, RL-based training, and evaluation of interactive systems.

3.The authors articulate their position clearly and develop it in a relatively systematic manner. Even though this is a position paper, the narrative is easy to follow. The paper first motivates the problem, then introduces its central abstractions, and then uses these concepts to reinterpret common agent failure modes and training implications. In particular, the “effort allocation” perspective provides a coherent organizing principle that connects internal reasoning, external acting, overthinking, and overacting under one lens.

4.The effort-allocation viewpoint is a useful conceptual contribution. The claim that tool use does not eliminate epistemic difficulty but instead reallocates where that difficulty is resolved is thought-provoking. This perspective gives the paper a stronger normative stance than work that only optimizes for correctness or reward, and it offers a language for discussing miscalibration even when final task success is high.

5.The paper engages with related work that is largely relevant and current. The cited references are mostly well matched to the paper’s topic, including work on overthinking, tool overuse mitigation, RL for reasoning or tool integration, workflow agents, and broader agent frameworks. The bibliography suggests that the authors are aware of contemporary discussions around agentic reasoning and tool use rather than positioning the work in isolation.

6.The discussion of training implications is relatively practical for a position paper. The final sections do not stop at conceptual claims. The authors attempt to connect their position to future training paradigms, including supervision, prompting, and reinforcement learning, and they explicitly discuss how agents might learn better calibrated tool-use behavior. This gives the paper a more constructive tone and shows that the authors are thinking beyond diagnosis toward possible solutions.

Weaknesses

1.The paper draws a fairly sharp opposition between internal knowledge and external tools, but this dichotomy may be too rigid in practice. In many realistic scenarios, tool use is not simply a fallback for cases where internal reasoning fails. External tools may be preferred because they are more precise, more efficient, safer, or better suited for verification. Likewise, external evidence may be used to support or calibrate internal knowledge rather than replace it. The paper would benefit from a more nuanced treatment of settings where internal reasoning and external tools are complementary rather than opposed.

2.Some of the formalization is lightweight relative to the strength of the claims built on top of it. The paper introduces equations and propositions to structure the discussion, but in several cases these mathematical expressions function more as conceptual categorization devices than as rigorous foundations. As a result, statements that these formulations provide a strong basis for evaluation or principle-driven alignment may feel somewhat overstated. This concern is especially relevant for the treatment of epistemic effort and its decomposition.

3.The paper makes many claims, but some of them are not supported with enough external evidence or detailed argumentation. The text contains several strong assertions, such as the long-term stagnation effect of unnecessary delegation, yet much of the support remains conceptual rather than empirical or deeply grounded in prior literature. The paper could be more convincing if, after introducing the core definitions, it connected each major proposition to more concrete evidence, case studies, or a broader set of supporting references.

4.The operational side of the framework remains underdeveloped. The paper acknowledges that key objects such as the true knowledge boundary and the minimal epistemic effort are latent. However, the practical consequences of this are substantial: if these quantities are not directly measurable, then it becomes unclear how strongly the framework can support evaluation, training, or diagnosis in real systems. The current discussion sometimes moves quickly from abstract definitions to normative conclusions without fully resolving this gap.

**Support:**

2

---

> ### Author Rebuttal · Authors · 2026-03-27
>
> We thank the reviewer for detailed feedback and recognition of six strengths. We address all items.
>
> `W1`: Internal and external are not opposites — §2.1 frames them as *alternative means of knowledge acquisition*. Figure 3(a) illustrates via segment AC: every point is a valid allocation. Points p (closer to A, lower α) and k (closer to C, higher α) represent ideal allocations under different preferences. Neither is a mistake if intended. Our normative concern is *unintentional drift toward A* from correctness-only training, suppressing internal capability development (Prop 3.8). Figure 3(b) handles tasks requiring structurally necessary external interaction (no Point C).
>
> `W2`: Prop 3.7 ($E^π_{int}+E^π_{ext}≥E*$) is retained as stated. Prop 3.8 will be reframed as a normative conjecture with the Appendix C.1 gradient-sparsity mechanism in the main text. Corroboration: SMART [1] shows tool overuse degrades performance; OTC-PO [2] confirms the contrapositive — penalizing delegation improves internal reasoning quality.
>
> `W3`: There are several evidence lines: (1) *Overthinking* [3,4]: RL length penalties reduce unnecessary tokens 65% on GSM8K, <2% accuracy loss. (2) *Overacting* [1,5]: SMART finds >30% unnecessary tool calls on internally-solvable tasks; OSWorld-Human shows top agents take 1.4–2.7× more steps than human-optimal. (3) *Cognitive offloading* [6,7]: reliance on external aids impedes internal skill development in humans — direct analogue of Prop 3.8; OTC-PO [2] reduces tool calls 68.3% while maintaining internal quality.
>
> `W4`: $p^{int}_t$ is estimable via rollout: (a) *Self-consistency*: K internal traces; high agreement → high solvability (Prop B.1). (b) *Draft reasoning + output confidence* as lightweight threshold. Calibrated self-awareness is what SFT/RL directly trains.
>
> `Q1`: (a) *Internal cognitive tools* [8]: The concept of cognitive tools has a well-established precedent in cognitive science [8], and cognitive offloading [6] formalizes the internal/external distinction our T_int/T_ext captures. (b) *Reasoning as acting*: both reasoning and tool-call tokens arise from the same autoregressive process; distinction is information provenance, not category. Advantages: generalizes ReAct to a principled decision problem; unified evaluation of overthinking/overacting as symmetric misallocations (Corollary 3.10).
>
> `Q2`: This is an initial study. α is *policy-dependent and varies at each step t* (Def 3.6 subscripts α to π). After a failed internal attempt, effective α decreases adaptively.
>
> `Q3`: Following RLVR: $E_{int}$ ≈ reasoning tokens; $E_{ext}$ ≈ tool calls; E^* ≈ minimum across correct rollouts (OTC-PO [2] for $E^*_{ext}$; efficient reasoning [4] for $E^*_{int}$). Prop 3.7's invariant is testable: methods reducing total effort must expand internal capability, not eliminate the epistemic requirement.
>
> `Q4`: See W1. Operating near A is not over-allocation if it is the intended preference (lower α). Over-allocation means unintentional systematic excess from miscalibrated training.
>
> `Q5`: See W3. Strongest validation: longitudinal comparison of RLVR-only vs. OTC-PO-trained models on held-out internally-solvable tasks; human analogue: cognitive offloading studies [6,7].
>
> `Q6`: Safety-critical domains warrant lower α — ideal point closer to A on segment AC, reflecting asymmetric error costs. This is not unnecessary delegation; ToA makes this preference explicit.
>
> `Q7`: Overthinking metrics [3,4]: unnecessary token count, token-accuracy curves, compute overhead. Overacting metrics [1,2,5]: tool overuse rate, step ratio vs. human-optimal, tool productivity, latency/cost. A calibrated agent scores well on all simultaneously.
>
> `Q8`: α is the tuning parameter: low α → closer to A (external-heavy); high α → closer to C (internal, capability development). ToA makes this tradeoff a principled design choice.
>
> `Q9`: (a) *SFT*: SMART [1] annotates steps as parametric vs. tool-dependent; reduces tool overuse 24%, improves performance 37%. (b) *RL*: OTC-PO [2] correctness-gated reward penalizes excess tool calls only among correct trajectories. (c) OSWorld-Human [5] human-optimal trajectories provide step-efficiency supervision.
>
> [1] SMART: Self-Aware Agent for Tool Overuse Mitigation, ACL Findings 2025
>
> [2] Teaching Language Model to Act Efficiently, NeurIPS LAW Workshop 2025
>
> [3] The Danger of Overthinking: Examining the Reasoning-Action Dilemma in Agentic Tasks, 2025
>
> [4] Training Language Models to Reason Efficiently, NeurIPS 2025
>
> [5] OSWorld-Human: Benchmarking the Efficiency of Computer-Use Agents, ICML Workshop on Computer-Use Agents 2025
>
> [6] Cognitive Offloading, Current Directions in Psychological Science, 2016
>
> [7] AI Tools in Society: Impacts on Cognitive Offloading and the Future of Critical Thinking, MDPI 2025
>
> [8] What are Cognitive Tools? 1992

---

> > ### Author Rebuttal · Reviewer_Vhxo · 2026-04-04
> >
> > Thanks for your detailed response. Overall, the current recommendation is appropriate.

---

### Decision · Program_Chairs · 2026-04-30

**Decision:**

Accept (regular)

**Comment:**

The overall review and the discussion suggest that the papers central idea is both timely and potentially useful for shaping future agent training and evaluation.  The rebuttal was strong: it addressed concerns about empirical grounding, operationalization, and formal positioning, and it fully resolved the most substantial conceptual objection. Remaining weaknesses mainly concern the latent nature of key variables and the fact that some claims remain more conceptual than directly validated, but this is acceptable for a position paper whose goal is to articulate a principled agenda rather than present a complete empirical system.